# Cases of Isolation of *Escherichia albertii* Strains from Commercial Quails with Gastroenteritis in Russia

**DOI:** 10.3390/microorganisms13040816

**Published:** 2025-04-03

**Authors:** Marat G. Teymurazov, Nikolay N. Kartsev, Alena A. Abaimova, Olga I. Tazina, Yuriy P. Skryabin, Olga E. Khokhlova

**Affiliations:** 1Federal Budget Institution of Science State Research Center for Applied Microbiology and Biotechnology (FBUN SRCAMB), Obolensk 142279, Russia; kartsev@obolensk.org (N.N.K.); jabress@mail.ru (A.A.A.); olgaglushkowa@mail.ru (O.I.T.); sjurikp@gmail.com (Y.P.S.); khokhlovaol@mail.ru (O.E.K.); 2Federal Scientific Center All-Russian Scientific Research Institute of Experimental Veterinary Sciences Named After K.I. Skryabin and Ya.R. Kovalenko of the Russian Academy of Sciences (FSBSI FSC VIEV RAS), Moscow 109428, Russia

**Keywords:** *Escherichia albertii*, *eae* genes, bird infections, APEC, antibiotic resistance

## Abstract

*Escherichia albertii* is a lactose-negative Escherichia that causes gastritis and enteritis in humans. An analysis of possible sources of infection points out that poultry may be a significant reservoir for this pathogen. The question of whether *E. albertii* can cause infections in poultry is still unanswered. Our article describes the isolation of *E. albertii*, for the first time in Russia, from the intestines of birds on a quail farm and a characterization of obtained cultures. We isolated different bacteria from pathological poultry material using bacteriological methods and ruled them out as probable causes for enteritis. The biochemical identification of *E. albertii* and antibiotic sensitivity were performed using a Vitek-2 Compact instrument. Bacterial identification was carried out using the MALDI-TOF Biotyper instrument. *E. albertii*-specific genes, virulence factor genes, and microcin genes were detected by real-time PCR. It was concluded that *E. albertii* isolated from sites of intestinal inflammation are a potential cause of enteritis and high poultry mortality—up to 15% of total livestock for 10- to 20-day-old quails. One of the *E. albertii* culture differed from the main group of *Escherichia* by its biochemical properties, and subsequent PCR analysis showed a lack of the intimin gene (*eae*). We describe the first occasion of infection caused by *E. albertii* in industrial quails. During the study, it was found that, according to the molecular–genetic and phenotypic properties of isolated strains in quails, there were at least two clonal groups of *E. albertii* differing in antibiotic resistance, biochemical indices, and presence of the *eae* (intimin) gene.

## 1. Introduction

*Escherichia albertii* is a Gram-negative, immobile, nonspore-forming, facultatively anaerobic, lactose-negative, bacillus-shaped bacterium belonging to the family *Enterobacteriaceae* [1]. *E. albertii* was first isolated in diarrhea cases in children from Bangladesh, tentatively misidentified as *Hafnia alvei*, and later classified as a new *Escherichia* species in 2003 [2]. *E. albertii* is an emerging enteropathogen in humans and many avian species [3]. This pathogen causes outbreaks of gastroenteritis, and some strains produce Shiga toxin.

Often, *E. albertii* is misidentified as enteropathogenic *E. coli* (EPEC) or enterohemorrhagic *E. coli* (EHEC) because of its genetic and phenotypic similarities to these pathogens. For example, *E. albertii* usually carries the *eae* gene encoding intimin, an important virulence factor also shared by pathogenic subgroups of *E. coli* [4]. This result probably led to an erroneous underestimation of the number of infections caused by *E. albertii*. For example, in several outbreaks of gastroenteritis, the causative agent was misdiagnosed as EPEC instead of *eae*-positive *E. albertii* [5].

The lack of a clear differential diagnosis remains a major problem, as there is no simple and accurate diagnostic protocol, especially for PCR typing, such as multilocus sequence typing (MLST) and O-genotyping [6,7].

To date, three biogroups of *E. albertii* have been distinguished according to their phenotypic features, as follows: biogroup 1—indole-negative, lysine-positive; biogroup 2—indole-positive, lysine-negative; biogroup 3—indole-positive, lysine-positive. This method may be useful for the identification of *E. albertii* in diagnostic laboratories or in the context of phylogeny [8].

Strains of *E. albertii* have been isolated from various animal sources, such as poultry, pigs, cats, dogs, bats, and raccoons, and raw meat of animal origin [8], but its natural reservoirs and routes of transmission to humans remain uncertain.

Nevertheless, an analysis of possible sources indicates that birds are the most significant reservoir for this pathogen [1]. Thus, the isolation of *E. albertii* cultures from migratory birds of the Asian–Australasian flyway (EAAF), which affects the territory of Russia, has been reported [9]. The genetic diversity of *E. albertii* isolates from migratory birds was demonstrated, and some isolates can potentially cause disease in humans. A large-scale study conducted on nine farms in Mississippi and Alabama showed that, of 270 cloacal swabs (30 per farm) tested by PCR diagnostics, 43 were positive for the presence of *E. albertii*, and 12 strains of *E. albertii* were isolated with varying degrees of isolation on individual farms from 0% to 23.3% [10]. Phylogenetically, isolates from different farms were distant but clonally similar from the same farm, including those with antibiotic and plasmid resistance. Importantly, 9 out of 12 *E. albertii* strains exhibited multiple drug resistance; 1 strain was even resistant to imipenem, a clinically important carbapenem antibiotic. In addition, comparative genomics analysis showed that two clusters of *E. albertii* strains isolated from chickens exhibited very close evolutionary relationships and similar virulence gene profiles to *E. albertii* strains isolated from humans.

A study in Switzerland on the detection of *E. albertii* in wild birds showed that, out of 280 fecal samples (collected from 26 bird species), *E. albertii* isolates were obtained from 11 bird species belonging to eight families. Four of the eleven species were waterfowl. The other seven species, including raptors and ravens, often inhabit agricultural lands [11]. In the same study, pooled fecal samples from 150 broiler flocks (more than one million birds) did not reveal the presence of *E. albertii* in any of the samples.

It should be noted that the review studies cited were aimed at detecting *E. albertii* carriage, but not at detecting infection caused by this bacterium. Whether *E. albertii* can cause infection in poultry remains an open question.

The zoonotic character of the disease suggests the importance and requirement of stricter *E. albertii* transmission control measures, and a deeper understanding of the mechanisms of the pathogenicity and the virulence potential of *E. albertii* is necessary; very few data are available, as most studies have been conducted with very few strains from limited geographical areas [3].

Our paper describes the isolation of *E. albertii* isolates, for the first time in the Moscow region of the Russian Federation, from the intestines of birds of a quail farm and the phenotypic and molecular genetic characterization of the obtained cultures. In addition, we show that *E. albertii* is the cause of lethal infection in birds. The study also aims to determine the causes of the increased mortality of quails at 10–20 days old with signs of gastrointestinal disorders and diarrhea. The listed provisions were the objective of our work.

## 2. Materials and Methods

### 2.1. Object of the Study

Live quails came from a quail farm in the Moscow region with 25 animals. Anamnesis indicated a significant loss of livestock (up to 15% of the total livestock) at the age of 10 to 20 days of life with signs of gastrointestinal tract damage. Keeping birds on the farm is cage-based. The cage housing of quails of three crosses—Japanese, Pharaoh, and Manchurian golden quail—was used. Quails were killed under chloroform anesthesia, after which they were isolated using a nutrient media.

### 2.2. Nutrient Media and Seeding

Using dactyloscopic smears (see below), quail parenchymatous organs were seeded on nutrient agar №1 (Nutrient Media, SRCAMB, Obolensk, Russia), Endo agar (Nutrient Media, SRCAMB, Obolensk, Russia), BHI agar (HiMedia, Laboratories Pvt. Ltd., Mumbai, Maharashtra, India) and the addition of 10 μg/mL NADH (AppliChem, Barcelona, Spain), 5% (w/mL), fresh yeast extract, 1% (wt./volume) glucose (AppliChem), and 0.001% (wt./volume) L cysteine (LLC NPP PanEco, MR, Moscow, Russia). To obtain parenchymatous smears, parenchymatous organs (liver, lungs, heart, spleen) were briefly cauterized with ethanol, the organ was excised, and the cut surface was pressed against agar several times. To obtain scrapings from the suborbital sinuses, the upper part of the beak was excised with sterile scissors and scraped with a 2 μL loop. The contents of the loops were seeded on Petri dishes with the above media. The trachea and oviducts were scraped with loops and the loop contents were seeded onto agar dishes. The intestinal contents were sown with a 10 µL loop on the agar surface and further spread on the surface of the medium with new loops using the Gold method to isolate colonies. *Clostridium perfringens* agar (HiMedia, Laboratories Pvt. Ltd., Mumbai, Maharashtra, India) supplemented with nutrients was used to isolate clostridia. The seeded Petri dishes were placed in an anaerostat (Schuett-biotec, Göttingen, Germany) and cultured under anaerobic conditions using anaerobic bags cultured at 42 °C for 24 h.

Cups of crops were also duplicated. Part of the dishes was placed in an anaerostat under microaerophilic conditions (with a candle) at 37 °C. The other part was cultured under anaerobic conditions, using anaerobic bags, at 42 °C for 24 h. We conducted a general bacteriological investigation with the seeding of the material on a complex of nutrient media for cultivation under different conditions. In addition to the commonly used media for culturing *Salmonella*, we used a temperature of 42 °C to create selective conditions (some other bacteria do not grow).

Culture typing: The grown colonies were evaluated culturally and morphologically and then analyzed by mass spectrometry.

### 2.3. Mass Spectrometric Analysis

Sample preparation was performed as previously described [12]. Protein extraction was performed according to the protocol proposed by Bruker (BrukerDaltonics, GmbH, Bremen, Germany). The results were summarized using the FlexAnalysis 3.4 software (Bruker Daltonics, GmbH, Bremen, Germany). For identification by direct protein profiling, an extended direct application method with formic acid was used, for which the biomass was applied in a uniform thin layer from an isolated colony of fresh culture into a target well using a needle or a disposable microbiological loop; after drying, 1 μL of 70% formic acid was applied to the biomass; after drying, 1 μL of the matrix solution was applied.

For protein extraction, one colony of the microorganism was transferred to an Eppendorf tube with 300 μL of water and vortexed for 1 min; 900 μL of 96% ethanol was added and shaken for 1 min, centrifuged for 2 min at 11,000× *g*; the supernatant was removed; the precipitate was air-dried, suspended in 20–80 μL of 70% aqueous formic acid solution, then an equivalent volume of 100% acetonitrile was added; 1 μL of the resulting supernatant was applied to the metal target of the mass spectrometer and dried at room temperature; 1 μL of matrix solution (a solution of α-cyano-4-hydroxycinnamic acid in an aqueous solution of 50% acetonitrile and 2.5% trifluoroacetic acid) was applied to the surface of the dried supernatant and dried at room temperature. For each studied unit (colony, strain), 5 wells were used to obtain a reliable result.

### 2.4. PCR Analysis

DNA was isolated from bacterial culture grown on nutrient agar dishes (HiMedia, Mumbai, India) using simple cell lysis. The total volume of the PCR reaction was 25 μL, including 5 μL of the DNA sample, 0.4 mM dNTP (Thermo Fisher Scientific, Waltham, MA, USA), 2.5 μL DreemTaq Green Buffer 10× (Thermo Fisher Scientific, USA), 10 mM of each primer, and 0.1 units/μL DreemTaq DNA polymerase (Thermo Fisher Scientific, USA). The amplification products were analyzed using the ChemiDoc XRS+ system (Bio-Rad, Hercules, CA, USA).

*E. albertii* strains were identified by searching for *E. albertii*-specific genes, such as *mdh*, *lysP*, and *clpX* (Table 1). We amplified the malate dehydrogenase gene (*mdh*), lysine-specific transporter gene (*lysP*), and heat shock protein gene (*clpX*) using multiplex PCR (Table 1) [13].

In addition, we used the real-time PCR differentiation of *E. coli* pathogroups (EPEC, ETEC, EIEC, EHEC, EAgEC) with the AmpliSens^®^ Escherichiose-FL kit (AmpliSens, Moscow, Russia) for isolated *E. albertii* and *E. coli* cultures.

The identification of the intimin gene, *eae* (common between *E. albertii* and EPEC), in *E. albertii* cultures was performed according to the European Union reference laboratory methodology for *E. coli* “Identification and characterization of verocytotoxin-producing *Escherichia coli* (VTEC) by real-time PCR amplification of major virulence genes and genes associated with serogroup mainly associated with severe human infections” [14].

### 2.5. CDT Typing

Cytolethal cell distension toxin (CDT) and cell cycle inhibitory factor (*cif*), types of cyclomodulins of pathogenic *E. coli* that are not associated with any specific phylogroup, were investigated using specific primers (Table 1) [15].

Virulence factor genes of APEC (avian pathogenic *E. coli*) were investigated according to the protocols proposed in the articles (Table 1) [16,17].

Microcin genes were determined according to the conditions proposed in [18].

**Table 1 microorganisms-13-00816-t001:** List of primers used in the study.

The *mdh*, *lysP*, and *clpX* Genes Are Specific to *E. albertii*					
Gene	Primer	5′-3′	Annealing, °C	Product Size (bp)	Reference
*Mdh* malatedehydrogenase	mdh-F	CTGGAAGGCGCAGATGTGGTACTGATT	55	115	[13]
	mdh-R	CTTGCTGAACCAGATTCTTCACAATACCG			
*Lys* Lysine-specific transporter	lysP-F	GGGCGCTGCTTTCATATATTCTT	55	252	[13]
	lysP-R	TCCAGATCCAACCGGGAGTATCAGGA			
*Clp* Heat shock protein	clpX-F	TGGCGTCGAGTTGGGCA	55	384	[13]
	clpX-R	TCCTGCTGCGGATGTTTACG			
Pre-denaturation 95 °C—5 min; 25 cycles: denaturation 95 °C—1 min, annealing 55 °C—1 min, elongation 72 °C—1 min. Final elongation 72 °C—3 min.					
***E. coli* APEC virulence genes**					
*cva* colicin V plasmid	cvaF	CACACACAAACGGGAGCTGTT	63	672	[16]
	cvaR	CTTCCGCAGCATAGTTCCAT			
*omp* episomal outer membrane protease	ompF	TCATCCCGGAAGCCTCCCTCACTACTAT	63	496	[17]
	ompR	TAGCGTTTGCTGCACTGGCTTCTGATAC			
*iroN* salmochelinsidero-phore receptor	ironF	AAGTCAAAGCAGGGGTTGCCCG	63	667	[16]
	ironR	GATCGCCGACATTAAGACGCAG			
*fim* fimbriae	fimF	GGATAAGCCGTGGCCGGTGG	63	331	[16]
	fimR	CTGCGGTTGTGCCGGAGAGG			
*iut* Aerobactinsidero-phore receptor	iutF	GGCTGGACATCATGGGAACTGG	63	302	[17]
	iutR	CGTCGGGAACGGGTAGAATCG			
*iss* Episomal gene for increased survival in serum	issF	CAGCAACCCGAACCACTTGATG	63	323	[16]
	issR	AGCATTGCCAGAGCGGCAGAA			
*hly* hemolysin	hlyF	GGCCACAGTCGTTTAGGGTGCTTACC	63	450	[17]
	hlyR	GGCGGTTTAGGCATTCCGATACTCAG			
*eae* intimin	eaeF	CATTGATCAGGATTTTTCTGGTGATA	63	102	[14]
	eaeR	CTCATGCGGAAATAGCCGTTA			
Pre-denaturation 95 °C—5 min, 35 cycles: denaturation 95 °C—30 s, annealing 63 °C—45 s, elongation 72 °C—1 min 45 s. Final elongation 72 °C—5 min.					
***E. coli* microcin genes**					
Microcin B17	mcc B17-F	TCACGCCAGTCTCCATTAGGTGTTGGCATT	60	135	[18]
	mcc B17-R	TTCCGCCGCTGCCACCGTTTCCACCACTAC			
Microcin C7	mcc C7-F	CGTTCAACTGTTGCAATGCT	60	134	
	mcc C7-R	AGTTGAGGGGCGTGTAATTG			
Microcin E492	mcc E492-F	GTCTCTCCTGCACCAAAAGC	60	291	
	mcc E492-R	TTTTCAGTCATGGCGTTCTG			
Microcin H47	mcc H47-F	CACTTTCATCCCTTCGGATTG	60	227	
	mcc H47-R	AGCTGAAGTCGCTGGCGCACCTCC			
Microcin J25	mcc J25-F	TCAGCCATAGAAAGATATAGGTGTACCAAT	60	175	
	mcc J25-R	TGATTAAGCATTTTCATTTTAATAAAGTGT			
Microcin L	mcc L-F	GGTAAATGATATATGAGAGAAATAACGTTA	60	233	
	mcc L-R	TTTCGCTGAGTTGGAATTTCCTGCTGCATC			
Microcin V	mcc V-F	CACACACAAAACGGGAGCTGTT	60	680	
	mcc V-R	TTTCGCTGAGTTGGAATTTCCTGCTGCATC			
Microcin M	micM-4-F	CGTTTATTAGCCCGGGATTT	60	166	
	micM-4-R	GCAGACGAAGAGGCACTTG			
Pre-denaturation 95 °C—5 min, 35 cycles: denaturation 95 °C—30 s, annealing 60 °C—30 s, elongation 72 °C—30 s. Final elongation 72 °C—5 min.					
**Primers encoding different types of *cdt* genes and *cif* gene in *E. coli***					
*cdtB-II*, *cdtB-III*, *cdtB-V*	CDT-s1	GAAAGTAAATGGAATATAAATGTCCG	60	467	[15]
	CDT-as1	AAATCACCAAGAATCATCCAGTTA			
*cdtB-II* *	CDT-IIas	TTTGTGTTGCCGCCGCTGGTGAAA	62	556	
*cdtB-III*, *cdtB-V* *	CDT-IIIas	TTTGTGTCGGTGCAGCAGGGAAAA	62	555	
*cdtB-I*, *cdtB-IV*	CDT-s2	GAAAATAAATGGAACACACATGTCCG	56	467	
	CDT-as2	AAATCTCCTGCAATCATCCAGTTA			
*cdtB-I*	CDT-Is	CAATAGTCGCCCACAGGA	56	411	
	CDT-Ias	ATAATCAAGAACACCACCAC			
*cdtB-IV*	CDT-IVs	CCTGATGGTTCAGGAGGCTGGTTC	56	350	
	CDT-IVas	TTGCTCCAGAATCTATACCT			
*cdtC-V*	P105	GTCAACGAACATTAGATTAT	49	748	
	c2767r	ATGGTCATGCTTTGTTATAT			
*cif*	cif-int-s	AACAGATGGCAACAGACTGG	55	383	
	cif-int-as	AGTCAATGCTTTATGCGTCAT			
Pre-denaturation 95 °C—5 min, 35 cycles: denaturation 95 °C—30 s, annealing—30 s, elongation 72 °C—30 s. Final elongation 72 °C—5 min.					

Note: * reverse primers used with the CDT-s1 primer.

### 2.6. Serotyping of Escherichia coli

The serotyping of *Escherichia coli* was performed according to the standard protocol recommended by the manufacturer (JSC “Biomed” named after. I.I. Mechnikov, Russian Federation).

### 2.7. Biochemical Identification and Susceptibility to Antimicrobials

Biochemical identification of *E. albertii* and antibiotic sensitivity were performed using GN cards on a Vitek-2 Compact instrument (BioMérieux, Marcy-l’Étoile, France).

An antimicrobial susceptibility testing was performed on the Vitek2 Compact system (BioMérieux, France) using the AST-N101 card (BioMérieux, France). A method of serial dilutions in microplates using antimicrobial chemotherapeutic agents (Sigma Aldrich, Merck, Burlington, MA, USA) was also used. The interpretation was made using the requirements of the European Committee on Antimicrobial Susceptibility Testing (EUCAST), version 12.0, 2022 (http://www.eucast.org) (accessed on 5 September 2022). *E. coli* ATCC 25922 strain was used as quality control.

*E. albertii* strains were deposited in the State Collection of Pathogenic Microorganisms under numbers SCPM-O-B-10824–SCPM-O-B-10835.

### 2.8. Statistical Analysis

Statistical processing was performed using SPSS, version 10.07. The method of variation statistics: arithmetic mean (M), its error (± m), 95% confidence interval of CI, Mann–Whitney criterion. Descriptive statistics: The results are presented as proportions with 95% confidence intervals for the MIC values. The calculation of the MIC and other indicators was carried out on the platform WHONET ver. 5.6. The results were considered reliable at *p* < 0.05. The qualitative features were proportions (%) and absolute numbers; in the comparative analysis, a two-sided Fisher criterion was used.

## 3. Results

### 3.1. Characteristics of E. albertii and Other Microorganisms

The mortality rate mortality of quails at 10–20 days old with signs of gastrointestinal disorders and diarrhea, depending on the workshop, ranged from 8% to 20%. Antibiotic therapy with a complex preparation including enrofloxacin and colistin before laboratory testing showed no obvious improvement in quail health.

According to the results of the study, most quails carried *Salmonella enterica* Bredeney, which is the causative agent of avian salmonellosis. *Salmonella* strains were not isolated from the gastrointestinal tract. There was respiratory salmonellosis, which was localized mainly in the trachea and suborbital sinuses. The identity of *Salmonella* was confirmed using PCR and agglutinating serum. Strains of *Streptococcus pluranimalium* causing avian streptococcosis, *Bordetella hinzii*, and *Gallibacterium anatis*, the causative agents of avian pasteurellosis (avian hemorrhagic septicemia)-like infection in birds, were isolated from the same localization.

At the same time, viral infections, such as avian influenza, infectious bronchitis, and Newcastle virus, were excluded by PCR analysis. In isolated strains of *Clostridium perfringens*, PCR analysis did not detect the NetB toxin, which has a leading role in the development of necrotizing enteritis, and the strains were classified as toxotype A according to the table proposed by Van Immersee [19].

*E. coli* strains isolated from the intestine were not assigned to any of the *Escherichia* pathogroups (EPEC, ETEC, EIEC, EHEC, EAgEC) by real-time PCR-RT, and only four isolated strains had genes that could be assigned to the APEC pathogroup. In total, 75% of strains of this pathogroup were isolated from the tracheas.

The intestine was found to have the following lesions: greenish feces, injected intestinal and mesenteric vessels, wall thickening, necrotic, and hemorrhagic and ulcerated areas (Figure 1). Strains were isolated from poultry with these features in one of the workshops and classified as *E. albertii* during the study. The results of the bacteriological examination are summarized in the Table 2.

### 3.2. Results of Identification of E. albertii and Antimicrobial Susceptibility

The strains isolated on Endo agar were lactose-negative and were initially considered to be *E. coli/Salmonella* spp., because no significant cultural and morphological differences from colonies of *E. coli* strains were detected, and the bird was a carrier of *Salmonella* spp. The mass spectral analysis did not provide an unambiguous interpretation, with close ID values for both *E. albertii* and *E. coli*.

Vitek2 analysis using GN maps showed that all strains were *E. coli* with a probability of more than 90% [95% CI = 88.7–98.9]. At the same time, it was determined that, conditionally, they should be divided into two groups according to the atypical features of *E. coli*. Strains with uncharacteristic features for *E. coli* in the first group included AGAL (alpha-galactosidase), BGAL (beta-galactosidase), and BGUR (beta-glucuronidase), and most strains, for the second group, included AGAL, BGAL, BGUR, and PHOS (phosphatase). Further characterization of antibiotic resistance using the Vitek2 device showed that they differed in antibiotic sensitivity as well (Table 3). Thus, culture–genetic differences were observed in strains of poultry from the same housing.

All strains of *E. coli* and *Salmonella* spp. were susceptible to beta-lactams, aminoglycosides, and tigecycline (Table 4).

The difference between the two clonal groups *E. albertii* was in their relationship to the antimicrobials ampicillin and amoxicillin/clavulonic acid. In the first group, the strains were sensitive to them, whereas in the second group, they were resistant. Otherwise, the antibiotic resistance profile coincided, and strains were resistant to fluoroquinolones (ciprofloxacin). All strains of *E. albertii* were susceptible to aminoglycosides, nitrofurans, sulfonamides, aminoglycosides, fosfomycin, and colistin. Given the widespread use of fluoroquinolones in poultry farming (including on the livestock of this farm), the resistance of cultures to these groups is natural.

### 3.3. Results of PCR Analysis

The AmpliSens^®^ Escherichiosis-FL PCR-RT kit showed that all isolated strains of *E. albertii* belonged to the EPEC (enteropathogenic *Escherichia*) pathogroup and contained the *eae* gene (encoding intimin), except for one strain, which we designated as group 2, because it also differed from the others in biochemical parameters and antibiotic resistance.

According to the multiplex PCR data for the presence of the *E. albertii*-specific genes *mdh*, *lysP*, and *clpX*, all tested strains had all three genes, and thus, it was confirmed that they belonged to the species *E. albertii*.

Following the O-serotyping of strains, all tested *E. albertii* strains reacted simultaneously with sera for O18 and O101.

Following CDT- and *cif*-typing, according to the results, both strains contained *cdtB*-II, *cdtB*-III, *cdtB*-V, *cdtB*-I, and *cdtB*-IV genes. The presence of *cdtB*-II, *cdtB*-III, and *cdtB*-V genes was confirmed by positive results for all three primer pairs—CDT-s1+CDT-as1, CDT-s1+CDT-IIas, and CDT-s1+CDT-IIIas (Table 1). The presence of the *cdtB*-I and *cdtB*-IV genes was confirmed by positive results only using the primer pair CDT-s2+CDT-as2 multiplex (*cdtB*-I, *cdtB*-IV). Specific primers for these types separately did not provide results. The presence of *cdtB*-I and *cdtB*-IV genes homologous to *E. coli* was doubtful and should be verified by full-genome sequencing. The *cdtC*-V and *cif* genes were not detected.

In addition, the obtained *E. albertii* strains were tested for the presence of genes specific to pathogenic APEC and microcin genes (Table 1).

According to the results of the study, genes of the APEC pathogenicity group, such as the fimbriae gene (*fim*) and episomal serum survival enhancement gene (*iss*), were detected in *E. albertii* strains of both groups. The microcin M4 and microcin B17 genes were also identified, which apparently increased the competitive advantage of *E. albertii* among closely related species.

## 4. Discussion

We examined quails from an industrial farm for bacterial infections to determine the causes of mass mortality based on signs of gastrointestinal tract damage. A significant proportion of quails (48% of the total number of quails examined) were carriers of salmonellosis caused by *Salmonella enterica* Bredeney (O:4 H:Lv 1,7). However, the localization of all isolated *Salmonella* cultures in the upper respiratory tract indicates that they are not the cause of gastrointestinal tract lesions in poultry. For industrial poultry, the respiratory tract is often the “entry gate” for *Salmonella*. Moreover, it is possible that the respiratory route of *Salmonella* infection in birds is preferable over the fecal–oral route [20,21].

Although, in the above studies, in experimental infections with *Salmonella* by intratracheal or aerosolized methods, during the course of time, *Salmonella* colonization also occurred in the cecum, but not in organs (except lungs). In our case, we observed the only respiratory carriage of *Salmonellae*. *S. enterica* Bredeney itself is not described as a causative agent of salmonellosis infection in birds, whereas the consumption of poultry meat contaminated with this serovariant can cause gastroenteritis outbreaks in humans [22]. Thus, we excluded *Salmonella* as a possible cause of poultry mortality in this case.

Other isolated bacteria of importance in the poultry industry include *Gallibacterium anatis*. This bacterium is widely spread in poultry at industrial poultry farms in the Russian Federation [23,24]. However, the cultures of *G. anatis* isolated by us, although they have the RTX-toxin gene (*GtxA*), the key virulence factor, are not hemolytic (do not belong to the *haemolytica* biovar), and it is known that hemolytic variants of *G. anatis* are capable of causing enteritis [25]. Therefore, *G. anatis* was not an etiologic agent of GI lesions in our case.

Another possible cause of enteritis is *E. coli*. However, this study determined that the isolated *E. coli* strains did not belong to any of the pathogroups leading to GI lesions. Avirulent *E. coli* strains and APEC pathogens were detected. It is known that APEC are extraintestinal *E. coli* with a leading intratracheal route of infection [26]. This was confirmed in our studies, as three of four cultures of the APEC pathogroup were isolated from the trachea (Table 2).

Thus, we hypothesized that *E. albertii* is a possible pathogen responsible for GI lesions in quail. Many cases of human gastroenteritis caused by *E. albertii* have been described, but information on avian diseases caused by this pathogen is scarce. For example, the authors [27] describe the death of wild birds of the Vesjurk family (*Carduelis flammea*) caused by *E. albertii*. According to their data, the cause of death was more than 8000 birds (out of those identified) in Alaska. Moreover, the described pathological changes are similar to our observations.

Previously, in the late 1990s, *E. albertii* was determined to be the cause of wild bird mortality in Scotland, but the authors incorrectly identified it as *E. coli* O86:K61 [28].

In the process of typing *E. albertii*, the biochemical properties of the cultures were determined. In general, the results showed that the differences in the biochemical properties of *E. coli/E. albertii* were characteristic of previously described *E. albertii* cultures [27]. At the same time, one isolated *E. albertii* culture exhibited a slight difference compared with the other cultures. Further analysis of antibiotic resistance and diagnosis using molecular genetic methods confirmed the heterogeneity of this culture with the others. Thus, it can be stated that there were two clonal groups of *E. albertii* in birds—containing and not containing the intimin *eae* gene. CDT-toxin genes were present in both groups, but no shigatoxin genes were detected. Apparently, the strain not carrying the intimin gene (and in the absence of shigatoxin genes) is avirulent in humans. The data of genomes have revealed that *E. albertii* carries a large collection of virulence-related genes, many of which are shared with other pathogens, mainly EPEC and EHEC; in addition to *LEE* (encoded type III section system), the *cdtB* gene belonging to the *cdtB*-II/III/V subtype group and the *paa* gene are common traits in most *E. albertii* strains isolated in different geographic regions [3]. Importantly, the presence of large self-transmissible plasmids, although not very common, could contribute to the diversification of *E. albertii* and, perhaps, further adaptation to different hosts [3]. The presence of the fimbriae (*fim*) and episomal serum enhanced survival (*iss*) genes may indicate the ability of these strains to survive and multiply in the bloodstream. In addition, the presence of these genes shows affinity to APEC strains, thus increasing pathogenicity, including for birds [3].

However, we have to perform model experiments, which we will conduct later, to prove that the bird is not just a reservoir for the pathogen but is also sick itself. For the remaining *E. albertii* strains, the virulence potential will be studied later after full-genome sequencing of the data is performed. In addition, it is necessary to obtain in vivo data on laboratory animals for a definitive conviction that they are virulent to birds, because the mechanisms of pathogenesis in humans and birds may be different [28]. We need studies of the mechanisms that regulate the expression and accumulation of virulence- and host adaptation-related genes. This work will be continued in the future.

It is interesting to note the presence of microcin M4 and B17 genes in the isolated cultures. The M4 microcin of class IIb (siderophore peptide) [29] has antimicrobial activity against *E. coli* and *Salmonellae*, which may explain the fact that *S. enterica* is not found in the intestine. B17 is a class I microcin, a post-translationally modified peptide that inhibits bacterial DNA gyrase [30] and has an even broader spectrum of antimicrobial activity affecting most *Enterobacteriaceae*. It is possible that the presence of these microcins may increase the distribution of *E. albertii* in the intestine, ultimately increasing the risk of infection caused by it.

Accordingly, large collections of strains from diverse regions, isolated from animals, humans, and various environments, need to be analyzed to have more general knowledge of *E. albertii* biology and their role in the pathology of animals and human.

## 5. Conclusions

The significant mortality of poultry on quail farms with signs of GI lesions is caused, in our opinion, by *E. albertii*. During the study, it was found that, according to the molecular–genetic and phenotypic properties of isolated strains in quails, there were at least two clonal groups of *E. albertii* differing in antibiotic resistance, biochemical indices, and presence of the *eae* (intimin) gene. CDT-toxin genes were present in all strains, and shigatoxin genes were absent. The described infection is the first evidence of *E. albertii* as a cause of death in industrial poultry.

## Figures and Tables

**Figure 1 microorganisms-13-00816-f001:**
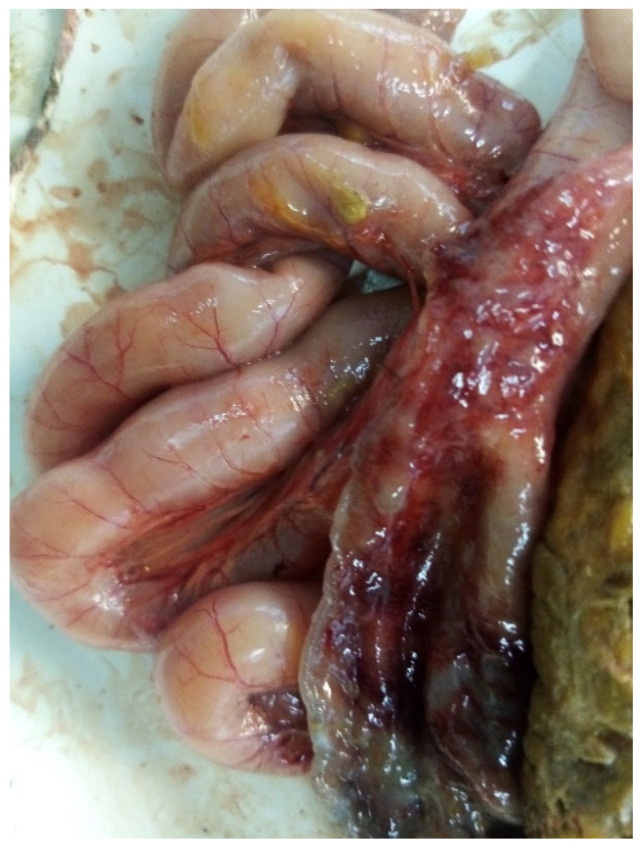
Pathological anatomical image of the dissection of the affected intestine with the site from which strains of *E. albertii* were isolated.

**Table 2 microorganisms-13-00816-t002:** Summary of the results of the bacteriological isolation of pathogenic microorganisms from poultry.

No. of Sample, Workshop, Age		Parenchymatous Organs (Liver, Lungs, Heart, Spleen)	The Contents of the Sinuses, Trachea	Intestines
1 line, 21 d	1	*Staphylococcus piscifermentans*	*Salmonella enterica* Bredeney, *Streptococcus pluranimalium*, *Bordetella hinzii*	*Clostridium perfringens*
	2	*S. piscifermentans*	*S. enterica* Bredeney, *B. hinzii*	*C. perfringens*
	3	*S. piscifermentans*	*S. enterica* Bredeney	*C. perfringens*
	4	*S. piscifermentans*	*S. enterica* Bredeney, *S. pluranimalium*, *B. hinzii*	
	5		*S. enterica* Bredeney, *B. hinzii*	
2 line, 21 d	1	*S. piscifermentans*	*S. enterica* Bredeney, *S. pluranimalium*	
	2	*S. piscifermentans*	*Gallibacterium anatis*, *S. piscifermentans*	*C. perfringens*
	3	*S. piscifermentans*	*S. pluranimalium*	
	4	*S. piscifermentans*	*G. anatis*, *S. piscifermentans*	
	5	*S. piscifermentans*	*G. anatis*, *S. piscifermentans*	*C. perfringens*
3 line, 18 d	1	*S. piscifermentans*	*G. anatis*, *S. pluranimalium*, *B. hinzii*	
	2		*S. enterica* Bredeney, *S. pluranimalium*, *B. hinzii*	
	3	*S. piscifermentans*	*S. enterica* Bredeney, *S. pluranimalium*, *B. hinzii*, *G. anatis*	
	4	*S. piscifermentans*	*S. pluranimalium*, *B. hinzii*	
	5		*G. anatis*, *S. enterica* Bredeney	
4 line, 18 d	1		*G. anatis*, *S. pluranimalium*, *E.coli*	
	2	*S. piscifermentans*	*G. anatis*, *S. pluranimalium*, *S. piscifermentans*	
	3		*G. anatis*	
	4	*S. piscifermentans*	*G. anatis*, *S. pluranimalium*, *S. piscifermentans*	
	5	*S. piscifermentans*	*S. piscifermentans*	*Escherichia albertii*
m/f, 18 d	1		*S. enterica* Bredeney	*E. albertii*, *C. perfringens*
	2	*S. piscifermentans*	*E.coli*	
	3		*E.coli*	*E. albertii*
	4		*S. enterica* Bredeney	*E. albertii*
	5		*S. enterica* Bredeney	*E. albertii*

**Table 3 microorganisms-13-00816-t003:** Antimicrobial susceptibility of *E. albertii* strains (MIC, mg/L) by Vitek2.

Antimicrobials	*E. albertii* Group 1 with the *eae* Gene (*N* = 11)		*E. albertii* Group 2 Without *eae* Gene (*N* = 1)	
	MIC, mg/L	Interpretation	MIC, mg/L	Interpretation
ESBL *	-		-	
Ampicillin	8 [100%; 95% CI]	S	16	R
Amoxicillin/clavulonic acid	8 [100%; 95% CI]	S	16	R
Cefotaxime	≤0.25 [100%; 95% CI]	S	≤0.25	S
Ceftazidime	≤0.12 [100%; 95% CI]	S	≤0.12	S
Cefipime	≤0.12 [100%; 95% CI]	S	≤0.12	S
Ertapenem	≤0.12 [100%; 95% CI]	S	≤0.12	S
Meropenem	≤0.25 [100%; 95% CI]	S	≤0.25	S
Amikacin	≤2 [100%; 95% CI]	S	≤2	S
Gentamicin	≤1 [100%; 95% CI]	S	≤1	S
Netilmicin	≤1 [100%; 95% CI]	ND	≤1	ND
Ciprofloxacin	1 [100%; 95% CI]	R	1	R
Tigecycline	≤0.5 [100%; 95% CI]	S	≤0.5	S
Fosfomycin	≤16 [100%; 95% CI]	S	≤16	S
Nitrofurantoin	≤16 [100%; 95% CI]	S	≤16	S
Trimethoprim/sulfamethoxazole	≤20 [100%; 95% CI]	S	≤20	S
Colistin	≤0.5 [100%; 95% CI]	S	≤0.5	S

Note: ESBL *—extended spectrum beta-lactamase; S—susceptibility; R—resistance.

**Table 4 microorganisms-13-00816-t004:** Antimicrobial susceptibility of *E. coli* and *Salmonella* spp. strains (MIC, mg/L) by method of serial dilutions in microplates.

Antimicrobials	*Salmonella* spp. (*N* = 12)		*E. coli* (*N* = 4)	
	MIC, mg/L	Interpretation	MIC, mg/L	Interpretation
Piperacillin	2 [100%; 95% CI]	S	2 [100%; 95% CI]	S
Piperacillin/tazobactame	2/4 [100%; 95% CI]	S	≤1/4 [100%; 95% CI]	S
Cefotaxime	0.25 [100%; 95% CI]	S	0.25 [100%; 95% CI]	S
Ceftazidime	2 [100%; 95% CI]	S	0.5 [100%; 95% CI]	S
Cefipime	≤0.12 [100%; 95% CI]	S	≤0.12 [100%; 95% CI]	S
Cefoperazone	≤0.5 [100%; 95% CI]	S	≤0.5 [100%; 95% CI]	S
Cefoperazone/sulbactam	≤0.5/0.25 [100%; 95% CI]	S	≤0.5/0.25 [100%; 95% CI]	S
Ertapenem	≤0.015 [100%; 95% CI]	S	≤0.015 [100%; 95% CI]	S
Meropenem	≤0.12 [100%; 95% CI]	S	≤0.12 [100%; 95% CI]	S
Netilmicin	1 [100%; 95% CI]	S	1 [100%; 95% CI]	S
Tobramycin	2 [100%; 95% CI]	S	0.5 [100%; 95% CI]	S
Tigecycline	0.25 [100%; 95% CI]	S	0.25 [100%; 95% CI]	S

Note: S—susceptibility.

## Data Availability

The original contributions presented in this study are included in the article. Further inquiries can be directed to the corresponding author.

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
