# Peer review of "Cases of Isolation of Escherichia albertii Strains from Commercial Quails with Gastroenteritis in Russia"

_microorganisms, 2025, doi:10.3390/microorganisms13040816_

Round 1
Reviewer 1 Report
Comments and Suggestions for Authors
microorganisms-3478045-peer-review-v1
Paper is interesting, reporting on detection of Escherichia albertii into samples from quails, farmed in Russian farming facility. Authors have selected a farm with high mortality and explored cases for this. Analyzing the samples, observations for the presence of different pathogens were recorded and focus been given to the strains of E. albertii. One interesting point is regarding whether authors have evidence that all isolates are in fact representative of just a few strains, or were they all different strains? I cannot record such as evaluation into the text. But this will be an important point to answer. Since maybe the same strain was spread into the farm and infected all animals. Well, at least 2 different strains were isolated and studied (2 groups based on the specific genes, positive and negative to the presence of that gene).
In my opinion the title can be formulated better. What characteristics? Resistance? Maybe you can be more specific. Furthermore, the word "industrial" maybe is not most appropriate. Maybe "commercially farmed quails in Russia"?
On my knowledge, the abstract do not need to be fractioned in subsections. Please, check the instructions from the journal and adjust.
Ln22: "detected by real-time PCR".
Ln25: Italics for Escherichia. Please, check the entire manuscript for adjustments of style and use of italics when appropriate.
In the abstract you need to focus on performed experiments and not on planned further experiments. The planned additional experiments can be stated at the end of the manuscript as further perspectives.
Maybe authors will consider reorganizing their abstract. In the current way is it repetitive and this can be avoid. Maybe help from more experienced colleges will be a good option to improve the writing quality of the manuscript. Moreover, improvement in the language needs to be made in several areas. Maybe professional linguists can help with this issue.
Ln37: In literature, often there is a confusion when the word "bacillus" was used. Maybe it will be appropriate to change to "bacillus shaped".
Ln39: Hafnia alvei - add iatlics.
Ln48: correct to: diagnostic.
Ln49: Remove "." before [5].
Ln78: Missing italics.
Introduction is informative, however, the biggest parts of this section can be moved to the discussion. In the introduction authors will need to keep only essential information about the Escherichia albertii and majority of the information regarding presence in the wild and farming animals/birds to be moved in the discussion.
Ln82: What part in Russian Federation farm was located. Russia is enormous country, and differences between west and east, north of south can be very significant. Thus, specify where in Russia farm was located will be a good addendum. Information is provided later in the material and methods section, but it will be positive if can be provided in this position as well.
Ln88: 25 animals? Is this correct number?
Ln91-93: This sentence is not clear, looks like something is missing.
Ln96 and further: For all applied material and equipment, please provide the address of the supplier, including name of the company, city, state (in case of federal countries) in abbreviated form and name of the country. Please, use headquarters and not local distributors. I presume that in mentioned example on Ln96 this is Russia, but information need to be provided. Also, what is the name of the company?
Ln104: What do you understand by "dense agar", please, provide % of agar supplement to the media.
Ln112-113: Please, explain why microaerophilic and anaerobic were performed at 37 and 42C, respectively.
Section 2.3. Please, provide a bit more details.
In section 2.4. something is not correct. If your total volume of the PCR reaction was 20ul, then 10x applied buffer need to be 2.0 ul, not 2.5 ul. Please, check the proportions of the PCR reaction and if needed, correct.
Table 1: Please, provide references
Table 2 and entire text: Please, consider using abbreviation for the bacterial names, after the full name was formally introduced.
Results are presented well, and informative. In fact, authors can consider improving their presentation. For example, provide figure 2 is not needed, since duplicates results presented in Table 3. It is not clear why authors have selected mentioned antibiotics to be tested. Maybe this needs to be clearly stated in material and methods where appropriate reference can be provided regarding this selection, and some additional arguments for the choice of the additional antibiotics.
Discussion needs to be presented better. Authors will need to focus on E. albertii, and all the results obtained in the previous section needs to be discussed, compared and analyzed regarding existing literature. Several virulence genes were explored into the manuscript, and it will be good if authors will discuss the importance of mentioned genes and fact that are present or not what consequences will bring to the studied E. albertii strains.
The conclusion can be more informative and maybe this will be the place where further plans for exploring the subject will need to be placed, and not into the abstract.
References needs to be adjusted. Some of them are not into the reused style.
Author Response
1. Summary Thank you very much for taking the time to review this manuscript. Please find the detailed responses below and the corresponding revisions/corrections highlighted/in track changes in the re-submitted files
3. Point-by-point response to Comments and Suggestions for Authors
Response: In my opinion the title can be formulated better. What characteristics? Resistance? Maybe you can be more specific. Furthermore, the word "industrial" maybe is not most appropriate. Maybe "commercially farmed quails in Russia"? Comments: thanks for the comment; we have changed the title of the article.
Response: On my knowledge, the abstract do not need to be fractioned in subsections. Please, check the instructions from the journal and adjust. Comments: we've redesigned the abstract
Response: Ln22: "detected by real-time PCR". |
Response : Ln25: Italics for Escherichia. Please, check the entire manuscript for adjustments of style and use of italics when appropriate. |
Comments : We have corrected the misspelling of microorganisms throughout the article.
Response : In the abstract you need to focus on performed experiments and not on planned further experiments. The planned additional experiments can be stated at the end of the manuscript as further perspectives. Maybe authors will consider reorganizing their abstract. In the current way is it repetitive and this can be avoid. Maybe help from more experienced colleges will be a good option to improve the writing quality of the manuscript. Moreover, improvement in the language needs to be made in several areas. Maybe professional linguists can help with this issue. Comments: Thank you for the comment. We tried to change the abstract - we removed the planned experiments and added specifics on the experiments conducted. As for the language, unfortunately we do not have a separate funding line for professional translation. Nevertheless, we have reworked the translation in several places to make the meaning of the text clearer.
Response: Ln37: In literature, often there is a confusion when the word "bacillus" was used. Maybe it will be appropriate to change to "bacillus shaped". Comments: Thanks for the comment - we've corrected it to "bacillus shaped". (Ln 39)
Response: Hafnia alvei - add iatlics. Comments : We've corrected it.
Response: Ln48: correct to: diagnostic. Comments 6: Thanks for the comment - we've corrected it (Ln 54) |
Response : Ln49: удалить «.» перед [5]. Comments: Thanks for the comment - we've corrected it
Response8: Ln78: Missing italics. Comments: Thanks for the comment - we've corrected it.
Response : Introduction is informative, however, the biggest parts of this section can be moved to the discussion. In the introduction authors will need to keep only essential information about the Escherichia albertii and majority of the information regarding presence in the wild and farming animals/birds to be moved in the discussion. Comments: Thank you. We have tried to modify the introduction.
Response : Ln82: What part in Russian Federation farm was located. Russia is enormous country, and differences between west and east, north of south can be very significant. Thus, specify where in Russia farm was located will be a good addendum. Information is provided later in the material and methods section, but it will be positive if can be provided in this position as well. Comments 10: Information is provided in the Materials and Methods section, In Ln99 the text is also contributed.
Response : Ln88: 25 animals? Is this correct number? Comments: Yes, that's correct.
Response: Ln91-93: This sentence is not clear, looks like something is missing. Comments: – Changes to the text have been made. Ln 110-113.
Response : Ln96 and further: For all applied material and equipment, please provide the address of the supplier, including name of the company, city, state (in case of federal countries) in abbreviated form and name of the country. Please, use headquarters and not local distributors. I presume that in mentioned example on Ln96 this is Russia, but information need to be provided. Also, what is the name of the company? Comments : Thanks for the comment - we've corrected it.
Response : Ln104: What do you understand by "dense agar", please, provide % of agar supplement to the media. Comments : These are the above commercial agars with exact formulation, may not have translated correctly, corrected. Ln123.
Response: Ln112-113: Please, explain why microaerophilic and anaerobic were performed at 37 and 42C, respectively. Comments : – We performed a general bacteriological investigation with sowing of material on a set of nutrient media for cultivation under different conditions. For example, for the isolation of Salmonellae, a temperature of 42C was used to create selective conditions (some other bacteria do not grow).
Response : Section 2.3. Please, provide a bit more details. Comment : – Changes have been made in section.
Response : In section 2.4. something is not correct. If your total volume of the PCR reaction was 20ul, then 10x applied buffer need to be 2.0 ul, not 2.5 ul. Please, check the proportions of the PCR reaction and if needed, correct. Comment : Thank you, the correct data has been entered.
Response : Table 2 and entire text: Please, consider using abbreviation for the bacterial names, after the full name was formally introduced. Comment: Introduced abbreviations for microorganisms. When first mentioned, the first name was retained to avoid confusion.
Response: Results are presented well, and informative. In fact, authors can consider improving their presentation. For example, provide figure 2 is not needed, since duplicates results presented in Table 3. It is not clear why authors have selected mentioned antibiotics to be tested. Maybe this needs to be clearly stated in material and methods where appropriate reference can be provided regarding this selection, and some additional arguments for the choice of the additional antibiotics. Comment: Thanks for the comments. Regarding the choice of antibiotics - with Vitek-2 we just focused on sensitivity to Enterobacteriaceae. As for the discs, we tried to be guided by CLSA and CASM-vet recommendations, but only for E.coli, as there are no recommendations for Escherichia albertii.
Response: Discussion needs to be presented better. Authors will need to focus on E. albertii, and all the results obtained in the previous section needs to be discussed, compared and analyzed regarding existing literature. Several virulence genes were explored into the manuscript, and it will be good if authors will discuss the importance of mentioned genes and fact that are present or not what consequences will bring to the studied E. albertii strains. The conclusion can be more informative and maybe this will be the place where further plans for exploring the subject will need to be placed, and not into the abstract. Comment: Thank you for your comments. We have tried to revise the conclusion, and plans for further research have been included in it. In fact, we are already writing a new article, which will be devoted to the annotation of E. albertii genomes, since we have carried out full-genome sequencing and are now placing the genomes in NCBI. In the same article, we also want to discuss virulence factors and other significant genes in detail. In the same article, we feel that it was important for us to show that E. albertii is the cause of poultry GI lesions and mortality against the background of isolation of other pathogenic bacteria for birds. Therefore, in the discussion we successively review the isolated bacteria and their potential as etiological agents of GI lesions and mortality in quail.
4. Response to Comments on the Quality of English Language |
Comment : Regarding language, unfortunately we do not have a separate funding line for professional translation. Nevertheless, we have reworked the translation in several places to make the meaning of the text clearer

Reviewer 2 Report
Comments and Suggestions for Authors
This is an interesting study on the isolation of E. albertii from quails, however, the research does not provide details on the pathogenesis of E. albertii infection. So the authors cannot conclude that E. albertii was the pathogen causing pathological changes and mortality. The authors had stressed their hypothesis, however, the conclusion should be based on the evidence.
Please explain the “industrial qual”.
How did the authors determine clonal groups of E. albertii?
Introduction.
Please provide comprehensive information about E. albertii infection in poultry including clinical symptoms and pathological findings. The paragraph describing the prevalence of E. albertii in wild birds )l.58-78) should be shortened.
L.57. Please provide a short description of the importance of the pathogens for human health.
Clearly distinguish between phenotypical and genotypical identification methods, and describe them separately.
The objectives of the study do not correspond to the title of the present manuscript.
Material and methods.
Please provide more detailed information about the farm, birds' housing and the outbreak. How many birds were selected for further analysis and what were the criteria for bird selection? What were the differential diagnoses for this infection and how were they excluded?
How was each agar evaluated? Which colonies were selected for further analysis? How many isolates were identified and used for DNA extraction?
How was the serotyping performed? Which antisera were used for analysis?
How were microorganisms prepared for AMR detection? Which breakpoints were used for the detection of AMR? What was the concentration of antimicrobial in disks?
Results.
The aim of the study does not correspond to the previously reported.
Could antibiotic therapy influence the results of AMR testing?
How was the mortality rate calculated?
Please check spelling of salmonella serovars. Italicize the genes.
What is a ‘respiratory salmonellosis?
L.169. The authors do not describe birds' necropsy or at least, the organs used for microbiological examination.
L.170. How was confirmation of Salmonella done? PCR and serotyping?
L.173. What is the same location?
L.174. How were those tests done?
L197. Were both Vitek2 or Maldi-tof used for the final confirmation of bacterial cultures? So based on which methods the final bacterial ID is provided?
L206. What are “culture-genetic” differences”?
Table 3. Please provide full names for all abbreviations.
L224. The PCR kit itself cannot provide the results.
The discussion
L251. What is a mass mortality?
L262. “bredeney serovariant”?
L 286-302 provides a general description of the present study design and future research direction. The weaknesses of the present study cannot become a central part of the manuscript.
English must be revised especially in terms of professional terminology.
Author Response
Thank you very much for taking the time to review this manuscript. Please find the detailed responses below and the corresponding revisions/corrections highlighted/in track changes in the re-submitted files.
Response 1: Please explain the “industrial qual” |
Comments 1: Means ‘commercial bird’ - we changed it in the text.
Response 2: How did the authors determine clonal groups of E. albertii? Comments 2: Based on the presence of the eae gene, profile of other genes, biochemical properties, and antimicrobial chemosensitivity profile. We have performed full genomic sequencing of total DNA of these strains, applied for their placement in GenBank, and plan to publish the results of analyses of the full genomic sequencing data in the next scientific publication. Already now, according to the data of the analysis of the whole-genome sequencing data, we have information that the strains of group 1 and group 2 belong to ST4633, but have differences in genome structures.
Response 3: Please provide comprehensive information about E. albertii infection in poultry including clinical symptoms and pathological findings. The paragraph describing the prevalence of E. albertii in wild birds )l.58-78) should be shortened. Comments 3: There is little information on poultry infection, those that are available are included in the review and in the discussion, including clinical symptoms and pathological data. |
Response 4: The paragraph describing the prevalence of E. albertii in wild birds) l.58-78) should be shortened.
Response 5: L.57. Please provide a short description of the importance of the pathogens for human health. Corrected in introduction and in disussion of article. Escherichia albertii is an emerging enteropathogen of humans and many avian species and animals. This pathogen causes outbreaks of gastroenteritis, and some strains produce Shiga toxin. The current knowledge of the phylogenic relationship of E. albertii with other Escherichia species and the biochemical and genetic properties of E. albertii, with particular emphasis on the repertoire of virulence factors and the mechanisms of pathogenicity, and we hope this provides a basis for future studies of this important emerging enteropathogen. Response 6: The objectives of the study do not correspond to the title of the present manuscript. Comments 6: We agree with the comment and have changed the title of the manuscript.
Response 7.Material and methods.
Response 8. How was each agar evaluated? Which colonies were selected for further analysis? How many isolates were identified and used for DNA extraction? How was the serotyping performed? Which antisera were used for analysis? Comments 8: We conducted a total bacteriological investigation with sowing of pathological material from birds on a set of nutrient media for cultivation in different conditions in order to isolate a possible spectrum of infectious agents in accordance with regulatory documents approved in the Russian Federation. Five colonies of bacteria most typical in morphology for each bacterial species were selected from each dish. Each colony was plated on a matrix for MALDI-TOF typing. PCR (described in Materials and Methods) was also used to confirm the strains belonged to the genus E. albertii. Belonging to the genus Salmonella was also confirmed by PCR analysis (test system for the genus ‘Vet-Factor’, MR, Russia), belonging to the serovar was determined using monoreceptor salmonellosis sera (‘Kurskaya Biofabrica’, Russia).
Response 9: How were microorganisms prepared for AMR detection? Which breakpoints were used for the detection of AMR? What was the concentration of antimicrobial in disks? Comments 9: Sample preparation for antimicrobial chemopreparations was carried out in accordance with EUCAST requirements and the Vitek-2 instrument instructions, and ATCC 25922 Escherichia coli strain was used as a control. Interpretation of antibiotic sensitivity results was performed according to EUCAST (described in Materials and Methods). The results of the disc-diffusion method were removed from this article due to the fact that methods based on MIC determination are more reliable.
Response 10: Could antibiotic therapy influence the results of AMR testing? Comments 10: yes, because with long-term antibiotic therapy, antibiotics increase selective pressure in bacterial populations, causing vulnerable bacteria to die, while increasing the percentage of resistant bacteria that continue to grow, as well as horizontal transfer of resistance genes.
Response 11: How was the mortality rate calculated? Comments 11: information was provided by the farm vet. Response 12: Please check spelling of salmonella serovars. Italicize the genes. Comments 12: Thank you for the comment, corrected.
Response 13: What is a ‘respiratory salmonellosis? Comments 13: – in the discussion, we give references that salmonellosis can be in the respiratory form.
Response 14: L.169. The authors do not describe birds' necropsy or at least, the organs used for microbiological examination. Comments 14 organs are entered in the text, post-mortem examination is described in the veterinary report: The Bioethics Commission of State Research Center for Applied Microbiology and Biotechnology approved the study and granted it consent waiver status (approval code:#VR-2023/6, approval date: October 27, 2023).
Response 15: How was confirmation of Salmonella done? PCR and serotyping? Comments 15: After MALDI-TOF staging, Salmonella genus affiliation was confirmed by PCR analysis (test system for the genus ‘Vet-Factor’, MO, Russia), serovar affiliation was determined using monoreceptor salmonellosis sera (‘Kurskaya Biofabrica’, Russia).
Response 16: L.173. What is the same location? Comments 16: the text has been corrected, it was not correct English
Response 17: L.174. How were those tests done? Comments 17: Newcastle disease, influenza, and infectious bronchitis were excluded using PCR test systems (all Vet-Factor, MO, Russia). Clostridium genotyping was performed using a proprietary PCR test system based on the primers presented in this article
Response 18: L197. Were both Vitek2 or MALDI-TOF used for the final confirmation of bacterial cultures? So based on which methods the final bacterial ID is provided? Comments 18: The strains were identified on the basis of biochemical properties in Vitek2 and protein profile in MALDI-TOF; these methods are phenotypic and do not allow for precise differentiation of E. albertii. Genetic methods are more accurate methods of identification; therefore, species affiliation was additionally determined on the basis of detection of specific genes for E. albertii species in PCR analysis. A more accurate method of determining species affiliation is full genomic sequencing; we have carried out full genomic sequencing of total DNA of these E. albertii strains, applied for their placement in GenBank and plan to publish the results of analyses of full genomic sequencing data in the next scientific publication.
Response 19: L206. What are “culture-genetic” differences”? Comments 19: By ‘’cultural‘’ we mean differences in antibioticgramme and biochemical properties. By ‘genetically’ we mean differences in the presence of the eae gene.
Response 20: L224. The PCR kit itself cannot provide the results. Comments 20: Thanks for the comment, corrected it in the text.
Response 21: L251. What is a mass mortality? Comments 21: Probably a bad translation. It means that the mortality rate of livestock on the farm reaches 15 %.
Response 22: L262. “bredeney serovariant”? Comments 22: Сhanged the text to mean Salmonella enterica serovar Bredeney.
Response 23: L 286-302 provides a general description of the present study design and future research direction. The weaknesses of the present study cannot become a central part of the manuscript. Comments 23: We show the presence of virulence genes that give E. albertii the ability to cause infection, and we describe two cases of bird mortality caused by E. albertii - we did not find any other literature on E. albertii infection in birds. However, in our opinion, the collected results are sufficient to put forward such a hypothesis (infection of birds caused by E. albertii). However, we have to perform model experiments, which we will conduct later, to prove that the bird is not just a reservoir for the pathogen but is also sick itself.
|
4. Response to Comments on the Quality of English Language |
Point 1: |
Comment 1: Regarding language, unfortunately we do not have a separate funding line for professional translation. Nevertheless, we have reworked the translation in several places to make the meaning of the text clearer. |

Reviewer 3 Report
Comments and Suggestions for Authors
Albertiella escherichia is a lactose-negative Escherichia coli that causes human gastritis and enteritis. This paper conducts a trace-back analysis suggesting that the source of infection may be poultry, describing the first isolation of Albertia coli from the intestinal tracts of birds in a quail farm in Russia, and identifying the obtained cultures. The writing of the article is relatively good, with strong logical coherence, and the results obtained are reliable. However, there are several important issues that need improvement, which the authors should carefully consider to meet the manuscript acceptance criteria. 1. The data presented in the last column of Table 1 under "Reference" are incorrect, necessitating correction by the authors, which is crucial. 2. It is not advisable to connect Figure 2 and Table 3, as this would weaken the logical flow of the article; it is recommended to rearrange this section. 3. The results section is somewhat sparse; it is suggested to integrate the results and discussion and enrich the content to enhance the overall quality of the study. 4. The logic in the abstract is poor; it is recommended to make reasonable modifications according to academic paper standards. 5. The experimental methods lack statistical data; it is essential to supplement this part.
Author Response
Thank you very much for taking the time to review this manuscript. Please find the detailed responses below and the corresponding revisions/corrections highlighted/in track changes in the re-submitted files.
Comments 1. The data presented in the last column of Table 1 under "Reference" are incorrect, necessitating correction by the authors, which is crucial. Response 1: Thank you! We corrected of Reference in the table 1.
Comments 2. It is not advisable to connect Figure 2 and Table 3, as this would weaken the logical flow of the article; it is recommended to rearrange this section. Response 2: Thank you! We corrected this part of article.
Comments 3. The results section is somewhat sparse; it is suggested to integrate the results and discussion and enrich the content to enhance the overall quality of the study. Response 3: Thank you! We corrected of the results and discussion of article..
Comments 4. The logic in the abstract is poor; it is recommended to make reasonable modifications according to academic paper standards. Response 4: Thank you! We corrected of the abstract.
Comments 5. The experimental methods lack statistical data; it is essential to supplement this part |
Response 5: Descriptive statistics - results are presented as proportions with 99% confidence intervals for the MIC values.
|

Reviewer 4 Report
Comments and Suggestions for Authors
The authors aimed to determine the causes of increased mortality in 10- to 20-day-old quails presenting with gastrointestinal disorders and diarrhea. They isolated several bacterial species, including Escherichia albertii, and stated that this is the first report of E. albertii isolation from bird intestines on a quail farm in Russia. Furthermore, they demonstrated that E. albertii can cause lethal infections in birds, which is expected.
Overall, the study presents a relevant investigation, and the methodology appears sound. However, the manuscript's grammar and structure impact readability, making it difficult to follow the progression of the study. Improving clarity and organization would significantly enhance the manuscript’s narrative.
Please perform antimicrobial susceptibility testing (AST) for all E. albertii, E. coli, and Salmonella Bredeney isolates using the Vitek 2 system. The AST results obtained via disk diffusion appear inaccurate based on my observations and interpretation. Additionally, include an ATCC strain as a quality control measure.
Move Table 1 to the supplementary material and ensure the references are correctly formatted.
The results section should not begin with "The study aim was to...". Instead, clearly state the study objective at the end of the introduction.
In Table 3, correct the antibiotic names and clarify what "BLRS" refers to.
There is no need to duplicate antibiotic susceptibility results obtained from the Vitek 2 system using disk diffusion. Additionally:
Colistin should not have been tested using disk diffusion.
The cost or charge of antibiotic disks should not be included.
The resistance discrepancies observed with cefepime, tigecycline, fosfomycin, and amikacin (resistant by disk diffusion but susceptible by Vitek 2) need further clarification.
Cefepime resistance (DD) should be reconsidered based on the overall resistance pattern.
The discussion is too brief and should be expanded to address strategies for preventing such infections in quail farms.
Compare the findings with similar reports in other animal species and human infections. Additionally, consider whether these strains have been detected in the environment and discuss potential reservoirs.
Include a "Strengths and Limitations" section at the end of the discussion section to contextualize the study’s contributions and constraints.
Strain typing is essential for this type of study. At a minimum all isolates, including Clostridium, Staphylococcus, Salmonella, E. coli, E. albertii, and other species should be analyzed using Bruker’s Fourier Transform Infrared (FT-IR) spectroscopy system to fulfill this requirement.
Comments on the Quality of English LanguageThe manuscript's grammar and structure impact readability, making it difficult to follow the progression of the study. Improving clarity and organization would significantly enhance the manuscript’s narrative.
Author Response
Thank you very much for taking the time to review this manuscript. Please find the detailed responses below and the corresponding revisions/corrections highlighted/in track changes in the re-submitted files.
Comments 1. Please perform antimicrobial susceptibility testing (AST) for all E. albertii, E. coli, and Salmonella Bredeney isolates using the Vitek 2 system. The AST results obtained via disk diffusion appear inaccurate based on my observations and interpretation. Additionally, include an ATCC strain as a quality control measure.
Response 1: Antimicrobial susceptibility of all strains of E. albertii we tested by Vitek 2 system. Antimicrobial susceptibility of all of E. coli, and Salmonella Bredeney isolates we tested by method of serial dilutions in microplates. We corrected of Vethods and Results.
Comments 2. Move Table 1 to the supplementary material and ensure the references are correctly formatted.
Response 2. We corrected.
Comments 3. The results section should not begin with "The study aim was to...". Instead, clearly state the study objective at the end of the introduction.
Response 3. We corrected.
Comments 4. In Table 3, correct the antibiotic names and clarify what "BLRS" refers to.
Response 4. Very sorry, it was mistake, correct ESBL. We corrected.
Comments 5. There is no need to duplicate antibiotic susceptibility results obtained from the Vitek 2 system using disk diffusion.
Response 5. Thank you! We agree that MIC methods more relevant compare with disk diffusion method. We deleted of results of disk diffusion method.
Comments 6. Additionally: Colistin should not have been tested using disk diffusion.
The cost or charge of antibiotic disks should not be included.
Response 6. Thank you! We deleted of results of disk diffusion method.
Comments 7. The resistance discrepancies observed with cefepime, tigecycline, fosfomycin, and amikacin (resistant by disk diffusion but susceptible by Vitek 2) need further clarification.
Cefepime resistance (DD) should be reconsidered based on the overall resistance pattern.
Response 7. Thank you! We deleted of results of disk diffusion method.
Comments 8. The discussion is too brief and should be expanded to address strategies for preventing such infections in quail farms.
Compare the findings with similar reports in other animal species and human infections. Additionally, consider whether these strains have been detected in the environment and discuss potential reservoirs.
Response 8. We have added information to the discussion.
Comments 9. Include a "Strengths and Limitations" section at the end of the discussion section to contextualize the study’s contributions and constraints.
Response 9. We have added Strengths and Limitations to the discussion.
Comments 10. Strain typing is essential for this type of study. At a minimum all isolates, including Clostridium, Staphylococcus, Salmonella, E. coli, E. albertii, and other species should be analyzed using Bruker’s Fourier Transform Infrared (FT-IR) spectroscopy system to fulfill this requirement.
Response 10. Strains were identified based on biochemical properties in Vitek2 and protein profile in MALDI-TOF, these methods are phenotypic and do not allow accurate differentiation of E. albertii. More accurate identification methods are genetic, therefore, species affiliation was additionally determined based on the detection of specific genes for the E. albertii species in PCR analysis. A more accurate method for determining species affiliation is whole-genome sequencing, we performed whole-genome sequencing of total DNA of these E. albertii strains, submitted an application for their placement in GenBank and plan to publish the results of the whole-genome sequencing data analysis in the next scientific publication.
Comments on the Quality of English Language
The manuscript's grammar and structure impact readability, making it difficult to follow the progression of the study. Improving clarity and organization would significantly enhance the manuscript’s narrative.

Round 2
Reviewer 1 Report
Comments and Suggestions for Authors
In my opinion authors have improved the manuscript and paper can be recommended for publication
Author Response
Comment: In my opinion authors have improved the manuscript and paper can be recommended for publication
Response: Thank you very much for your help and recommendations!
Reviewer 3 Report
Comments and Suggestions for Authors
The author did not mark the revised sections in the latest manuscript, and did not address the fifth question I raised in the manuscript either. It seems that the author did not understand the question, which pertains to the need for a detailed description of the data processing and statistical methods in the Materials and Methods section of the paper.
Author Response
Thank you very much for taking the time to review this manuscript. Thank you very much for help and your comments! Please find the detailed responses below and the corresponding revisions/corrections highlighted/in track changes in the re-submitted files.
Comments 1. The author did not mark the revised sections in the latest manuscript.
Response 1: Thank you! All corrections in the article mark yellow.
Comments 2. and did not address the fifth question I raised in the manuscript either. It seems that the author did not understand the question, which pertains to the need for a detailed description of the data processing and statistical methods in the Materials and Methods section of the paper.
Response 2: Thank you! We have included item 2.8. Statistical analysis in Materials and Methods in the article – lines 190-197.

Reviewer 4 Report
Comments and Suggestions for Authors
The authors have addressed almost all the comments, and the manuscript is suitable for publication in its current form.
Author Response
Comments 1: The authors have addressed almost all the comments, and the manuscript is suitable for publication in its current form
Response 1: Thank you very much for your help and recommendations!
Round 3
Reviewer 3 Report
Comments and Suggestions for Authors
I would like to thank the author for his careful revision of the manuscript, which I believe is acceptable.